# Safety and impact of eculizumab withdrawal in patients with atypical haemolytic uraemic syndrome: protocol for a multicentre, open-label, prospective, single-arm study

Sarah Dunn [ORCID],[1] Victoria Brocklebank,[2] Andrew Bryant,[3] Sonya Carnell,[1] Thomas J Chadwick,[3] Sally Johnson,[2,4] David Kavanagh,[2,5] Jan Lecouturier,[3] Michal Malina [ORCID],[2,4] Eoin Moloney,[6] Yemi Oluboyede,[3] Christopher Weetman,[1] Edwin Kwan Soon Wong,[2,5] Len Woodward,[7] Neil Sheerin[2,5]

For numbered affiliations see end of article.

**Correspondence to**
Sarah Dunn;
sarah.dunn2@ncl.ac.uk

## ABSTRACT

**Introduction** Atypical haemolytic uraemic syndrome (aHUS) is a rare, life-threatening disease caused by excessive activation of part of the immune system called complement. Eculizumab is an effective treatment, controlling aHUS in 90% of patients. Due to the risk of relapse, lifelong treatment is currently recommended. Eculizumab treatment is not without problems, foremost being the risk of severe meningococcal infection, the burden of biweekly intravenous injections and the high cost.

This paper describes the design of the Stopping Eculizumab Treatment Safely in aHUS trial that aims to establish whether a safety monitoring protocol, including the reintroduction of eculizumab for those who relapse, could be a safe, alternative treatment strategy for patients with aHUS.

**Methods and analysis** This is a multicentre, non-randomised, open-label study of eculizumab withdrawal with continuous monitoring of thrombotic microangiopathy-related serious adverse events using the Bayes factor single-arm design. 30 patients will be recruited to withdraw from eculizumab and have regular blood and urine tests for 24 months, to monitor for disease activity. If relapse occurs, treatment will be restarted within 24 hours of presentation. 20 patients will remain on treatment and complete health economic questionnaires only. An embedded qualitative study will explore the views of participants.

**Ethics and dissemination** A favourable ethical opinion and approval was obtained from the North East-Tyne & Wear South Research Ethics Committee. Outcomes will be disseminated via peer-reviewed articles and conference presentations.

**Trial registration number** EudraCT number: 2017-003916-37 and ISRCTN number: ISRCTN17503205.

### STRENGTHS AND LIMITATIONS OF THIS STUDY

⇒ This is the first UK trial to evaluate the safety of eculizumab withdrawal in patients with atypical haemolytic uraemic syndrome (aHUS).
⇒ This trial fulfils the National Institute for Health and Care Excellence recommendation that a research programme, with robust methods, should be carried out to evaluate when stopping eculizumab treatment or dose adjustment might occur.
⇒ Clinical experience suggests that if relapse occurs, this will likely happen in the first 12 months of withdrawal; however, this trial follows patients up for 24 months to capture those patients who may relapse after the 12-month point.
⇒ The small number of patients with aHUS on treatment in the UK is insufficient to conduct a standard parallel group, randomised controlled trial.
⇒ COVID-19 has had an impact on recruitment.

characterised by thrombocytopenia, microangiopathic haemolytic anaemia and acute kidney injury (AKI), and other organ involvement. Historically, it is associated with a poor prognosis, with 50% of patients developing end-stage kidney disease or dying in the first year after presentation[1] and a high risk of disease recurrence after kidney transplantation.[2] Prior to 2011, treatment options were limited and relied on plasma infusion or exchange, but in many cases this treatment failed to influence the course of disease[1] and was itself associated with significant morbidity and mortality.[3] In the UK, the incidence of aHUS is 0.4–0.5 cases per million per year.[4]

The complement system is part of the innate immune system and in healthy individuals is tightly regulated to prevent excessive activation. In 60%–70% of patients with aHUS, a

## INTRODUCTION

Atypical haemolytic uraemic syndrome (aHUS) is a severe, life-threatening disease

genetic variant or autoantibody-increasing complement activation can be identified.[5] In these patients, excessive activation of complement leads to endothelial injury and thrombus formation. The underlying genetic variant that predisposes to disease has an influence on the severity of disease and the likelihood of recurrent disease developing after transplantation.[6]

Eculizumab is a humanised monoclonal antibody that inhibits the function of C5, an important protein involved in complement activation. Two uncontrolled, open-label trials involving 36 adult and adolescent patients demonstrated the efficacy of eculizumab treatment for aHUS over a 26-week period.[7] Additional prospective studies in children[8] and adults[9] confirmed efficacy. Follow-up of the original cohort suggests that treatment for 2 years is associated with good, longer-term clinical outcomes.[10] On the basis of the initial trial results, eculizumab was approved for the treatment of aHUS by the European Medicines Agency[5] and US Food and Drug Administration[11] in 2011. The National Institute for Health and Care Excellence (NICE) published its evaluation in 2015, recommending that eculizumab should be used for the treatment of aHUS.[12] A recommendation in the NICE evaluation was that funding was on the condition that there was a 'research programme with robust methods to evaluate when stopping treatment or dose adjustment might occur'.

Complement is part of the immune system, therefore, treatment with eculizumab is immunosuppressive, in particular, increasing the risk of *Neisseria meningitidis* infection (1–2000-fold).[13] [14] All patients are vaccinated against meningococcal infection before starting eculizumab and in the UK continuous prophylactic antibiotics are recommended. Despite these recommendations, there have been six cases of meningococcal infection in the UK in patients on eculizumab treatment for aHUS. We are also aware of uncommon infections occurring in this group (enteroviral pneumonitis and herpes simplex meningitis), but whether these are attributable to eculizumab treatment is unclear.

Although there have been reports of patients relapsing after the withdrawal of treatment,[15] there is increasing evidence that continuous treatment is not required for all patients. From experience prior to the introduction of eculizumab, the risk of relapse is greatest in the period immediately after first presentation with 82% of relapses in adults, and 57% of relapses in children occurring within the first year after disease onset. Beyond the first year, only a further 20% of patients will relapse in the subsequent 5–10 years.[1] Therefore, a proportion of patients will not relapse after their initial presentation and will be on eculizumab unnecessarily. In addition, with monitoring for relapse and early reintroduction of treatment, complications from relapse can be avoided.[16–18]

In this trial, we will test the safety of eculizumab withdrawal using a Bayesian trial design. The efficacy of self-monitoring will also be tested, and we will explore patients' and parents/legal guardians' views on how treatment and monitoring of disease can be delivered most effectively.

## METHODS AND ANALYSIS
### Objectives and outcome measures
#### Primary
The primary clinical objective is to determine the safety of eculizumab withdrawal in patients with aHUS, measured by the number of patients with a thrombotic microangiopathy (TMA)-related serious adverse event (SAE) defined as any of the following:
- Irreversible (>3 months) reduction in estimated glomerular filtration rate (eGFR) not attributable to another cause:
  - In adults:
    - By ≥20% if the screening eGFR is <90 mL/min/1.73 m$^2$.
    - By >20% to a level <90 mL/min/1.73 m$^2$ if the screening eGFR is >90 mL/min/1.73 m$^2$.
  - In children:
    - By ≥20% if the screening eGFR is <75 mL/min/1.73 m$^2$.
    - By >20% to a level <75 mL/min/1.73 m$^2$ if the screening eGFR is >75 mL/min/1.73 m$^2$.
- An episode of AKI attributed to a TMA that requires renal replacement therapy.
- A non-renal manifestation of a TMA that requires hospitalisation, causes irreversible organ damage or death.

#### Secondary
##### Clinical
1. Measure the effectiveness of a monitoring protocol to detect disease relapse following withdrawal of eculizumab as assessed by:
   a. The proportion of patients who relapse and restart eculizumab without the development of a TMA-related SAE.
   b. The time from the first clinical feature (symptom, positive urinalysis or laboratory result) of a relapse of TMA and the reintroduction of eculizumab.
2. The relapse rate after withdrawal of eculizumab as determined by the proportion of patients who relapse after eculizumab is withdrawn.
3. The proportion of patients, currently on long-term treatment with eculizumab, who can be maintained off treatment.
4. The period from withdrawal to relapse in those patients who restart treatment.
5. The change in estimated GFR as calculated by the Chronic Kidney Disease (CKD) Epidemiology Collaboration or modified Schwartz equations over the course of the study from baseline (day 0) to end of the study.
6. Important clinical and laboratory indicators of imminent relapse by review of reported symptoms, physical signs, urinalysis and laboratory results prior to the diagnosis of a relapse.

## Health economics

7. The costs and health outcomes (measured in terms of adverse events (AEs) and quality-adjusted life years (QALYs)) for patients on standard care (not withdrawing from eculizumab treatment) over the 2-year trial duration.
   a. Healthcare utilisation questionnaires for non-withdrawal participants at days 0, 14, 70, 154, 252, 336, 504 and 672.
   b. AE assessment at every visit from day 7 (32 visits) for withdrawal participants.
8. QALYs estimated from responses to the EuroQol - 5 Dimension - 5 Level (EQ-5D-5L) and Short Form Health Survey -36 item (SF-36), and determinants of QALYs/utilities over the 24-month follow-up period at days 0, 14, 70, 154, 252, 336, 504 and 672.
9. Model-based estimate of the costs and health consequences, with results presented in terms of cost per QALY gained, over the estimated lifetime of patients withdrawing from treatment compared with standard care.

### Trial design

This is a multicentre, non-randomised, open-label study of eculizumab withdrawal with continuous monitoring of TMA-related SAEs using the Bayes factor single-arm design of Johnson and Cook.[19] The patients will self-select whether they wish to withdraw from eculizumab and carry out the monitoring protocol or remain on treatment and be part of the health economics analysis only. An economics analysis, informed by the results of this trial, will determine whether eculizumab withdrawal, substituting treatment with a protocolised surveillance and treatment reintroduction strategy, is cost-effective. The patient visit schedule for the withdrawal cohort is shown in figure 1, and the health economics cohort in figure 2.

| Procedure | Central Pre-Screen | Site Screen and Consent | Final Infusion | V2 | V3 | V4 | V5 | V6 | V7 | V8 | V9 | V10 | V11 | V12 | V13 | V14 | V15 | V16 | V17 | V18 | V19 | V20 | V21 | V22 | V23 | V24 | V25 | V26 | V27 | V28 | V29 | V30 | V31 | V32 | V33 | V34 | Unscheduled Visit |
|---|---|---|---|---|---|---|---|---|---|---|---|---|---|---|---|---|---|---|---|---|---|---|---|---|---|---|---|---|---|---|---|---|---|---|---|---|---|
| Month | 0 | 0 | 0 | 1 | 1 | 1 | 1 | 1 | 2 | 2 | 3 | 3 | 4 | 4 | 5 | 5 | 6 | 6 | 7 | 8 | 9 | 10 | 11 | 12 | 13 | 14 | 15 | 16 | 17 | 18 | 19 | 20 | 21 | 22 | 23 | 24 | |
| Visit Number | N/A | N/A | 1 | 2 | 3 | 4 | 5 | 6 | 7 | 8 | 9 | 10 | 11 | 12 | 13 | 14 | 15 | 16 | 17 | 18 | 19 | 20 | 21 | 22 | 23 | 24 | 25 | 26 | 27 | 28 | 29 | 30 | 31 | 32 | 33 | 34 | |
| Genetic Eligibility | X | | | | | | | | | | | | | | | | | | | | | | | | | | | | | | | | | | | | |
| Medical History Review | | X | | | | | | | | | | | | | | | | | | | | | | | | | | | | | | | | | | | |
| Informed consent | | X | | | | | | | | | | | | | | | | | | | | | | | | | | | | | | | | | | | |
| Eligibility Checklist Completion | | X | | | | | | | | | | | | | | | | | | | | | | | | | | | | | | | | | | | |
| Physical Examination | | X | | | | | | | | | | | | | | | | | | | | | | X | | | | | | | | | | | X | | X |
| Height & Weight | | X | | | | | | | | | | | | | | | | | | | | | | | | | | | | | | | | | | | |
| Pregnancy test | | X | | | | | | | | | | | | | | | | | | | | | | | | | | | | | | | | | | | |
| Eculizumab Infusion | | | X | | | | | | | | | | | | | | | | | | | | | | | | | | | | | | | | | | |
| meningococcal prophylaxis | | X | | X | | | | | | | | | | | | | | | | | | | | | | | | | | | | | | | | | |
| urine analysis training | | | | X | | | | | | | | | | | | | | | | | | | | | | | | | | | | | | | | | |
| Vital Signs | | X | | X | X | X | X | X | X | X | X | X | X | X | X | X | X | X | X | X | X | X | X | X | X | X | X | X | X | X | X | X | X | X | X | X | X |
| Concomitant medication Review | | X | | X | X | X | X | X | X | X | X | X | X | X | X | X | X | X | X | X | X | X | X | X | X | X | X | X | X | X | X | X | X | X | X | X | X |
| Renal Function (Creatinine & GFR) | | X | | X | X | X | X | X | X | X | X | X | X | X | X | X | X | X | X | X | X | X | X | X | X | X | X | X | X | X | X | X | X | X | X | X | X |
| Liver Function Tests | | X | | X | X | X | X | X | X | X | X | X | X | X | X | X | X | X | X | X | X | X | X | X | X | X | X | X | X | X | X | X | X | X | X | X | X |
| Haemolysis markers (full blood count & LDH) | | X | | X | X | X | X | X | X | X | X | X | X | X | X | X | X | X | X | X | X | X | X | X | X | X | X | X | X | X | X | X | X | X | X | X | X |
| Electrolyte profile (U&Es) | | X | | X | X | X | X | X | X | X | X | X | X | X | X | X | X | X | X | X | X | X | X | X | X | X | X | X | X | X | X | X | X | X | X | X | X |
| Haptoglobin & Blood film | | X | | X | | | | X | | | X | | | | | | X | | | | | | | X | | | | | | X | | | | | | X | X |
| Urine PCR | | X | | X | | | | X | | | X | | | | | | X | | | | | | | X | | | | | | X | | | | | | X | X |
| Biomarkers and complement activation sample | | X | | X | X | X | | X | | X | | X | | | | | X | | | | X | | | X | | | | | | X | | | | | | X | X |
| Home Urinalysis | | X | | *Daily (Month 1)* | | | | | *3 times per week* | | | | | | | | | | | | | | | | | | | | | | | | | | | | |
| Home Urinalysis diary review | | | | X | X | X | X | X | X | X | X | X | X | X | X | X | X | X | X | X | X | X | X | X | X | X | X | X | X | X | X | X | X | X | X | | |
| Adverse Events | | | | X | X | X | X | X | X | X | X | X | X | X | X | X | X | X | X | X | X | X | X | X | X | X | X | X | X | X | X | X | X | X | X | X | X |
| EQ5D & SF36 | | | | X | | X | | | | | | X | | | | | X | | | | | X | | X | | | | | | X | | | | | | | X | |
| Health care Utilisation questionnaire | | | | X | | X | | | | | | X | | | | | X | | | | | X | | X | | | | | | X | | | | | | | X | |
| Time & Travel Questionnaire | | | | | | | | | | | | | | | | | | | | | | | | X | | | | | | | | | | | | | |

**Figure 1** Data collection time points for withdrawal cohort. GFR, glomerular filtration rate; LDH, lactate dehydrogenase; U&Es, urea and electrolytes.

| | Follow - up | | | | | | |
|---|---|---|---|---|---|---|---|
| month | 1 | 3 | 6 | 9 | 12 | 18 | 24 |
| visit number | 1 | 2 | 3 | 4 | 5 | 6 | 7 | 8 |
| Medical History Review | X | | | | | | | |
| Informed consent | X | | | | | | | |
| Eligibility Checklist | X | | | | | | | |
| EQ5D & SF36 | X | X | X | X | X | X | X | X |
| Health care Utilisation questionnaire | X | X | X | X | X | X | X | X |
| Time & Travel Questionnaire | | | | | | | X | |

**Figure 2** Data collection time points for health economics cohort.

## Trial setting

This multicentre trial will be carried out in up to 20 adult and paediatric renal units (secondary and tertiary care) in the UK that are using eculizumab to treat patients with aHUS.

## Eligibility

All patients must fulfil the following inclusion criteria in order to be eligible for the trial:
▶ Age ≥2+ years of age.
▶ On eculizumab treatment for at least 6 months.
▶ In remission with no evidence of ongoing microangiopathic haemolytic anaemia activity at screening defined by:
  Platelet count >lower limit of normal as determined by local reference range.
  Lactate dehydrogenase (LDH) <×2 upper limit of normal as determined by local laboratory reference ranges.
▶ Normal renal function or CKD stages 1–3 (eGFR >30 mL/min/1.73 m$^2$).
▶ Absence of decline of renal function confirmed by review of available assessments of renal function for the preceding 6 months by the chief investigator (CI) and clinical members of the Trial Management Group (TMG).

The following inclusion criteria must be met only by those wishing to participate in the withdrawal component of the trial:
▶ Willing to attend for safety monitoring assessments.
▶ Willing to travel only to countries that can supply eculizumab (to be confirmed with coordinating centre prior to travel).
▶ Able to perform or parent/guardian to perform and record self-monitoring urinalysis.
▶ Sexually active female patients must have a negative pregnancy test at screening and be using an effective contraception for the duration of the study.
OR
▶ Fulfil one of the following criteria:
  Be post-menopausal or have undergone surgical sterilisation.

The following exclusion criteria are applicable to all patients wishing to participate in the trial:
▶ Severe non-renal disease manifestations at initial presentation with aHUS, which in the opinion of the CI and/or the clinical members of the TMG makes the risk of treatment withdrawal unacceptable.
▶ Current or planned pregnancy within the study duration.
▶ Unable to give informed consent or assent, or unable to obtain parent/guardian consent if under 16 years of age.
▶ Current participation in another clinical trial (not including participation in aHUS registries).
▶ Severe, uncontrolled hypertension (systolic blood pressure >160 mm Hg) that is likely to induce at TMA.

The following exclusion criteria are applicable only to those wishing to participate in the withdrawal component of the trial:
▶ Loss of a previous transplant kidney to recurrent aHUS.
▶ Transplant recipient with a pathogenic mutation in *C3*, *CFH* or *CFB*.
▶ Haematuria rating of 3+.

## Screening and recruitment

Thirty patients will be recruited to withdraw from eculizumab treatment, and 20 patients will be recruited who will remain on treatment and complete the health economics questionnaires only.

Patients with a diagnosis of aHUS receiving eculizumab[4] to treat disease in native or transplanted kidneys will be identified by the National aHUS Service, which maintains a list of patients who fulfil these criteria as part of the National Health Service (NHS) England-commissioned service. Those patients who meet the genetic eligibility criteria will be highlighted to site teams who will carry out formal screening assessments. A physical examination and vital signs will be performed, and routine safety laboratory tests will be reviewed to ensure that a patient fulfils all eligibility criteria for entry into the study. Female participants withdrawing from treatment, who are of childbearing age and sexually active, will be required to have a negative pregnancy test prior to treatment withdrawal. Participants will also consent to have samples taken for exploratory analysis and storage at Newcastle University biobank for use in future research.

Consent will be sought from the parents/legal guardian on behalf of patients under the age of 16 years. Assent will be taken from those patients under 16 years old, as appropriate (online supplemental file 1). No trial-related procedures will be carried out prior to consent.

## Intervention

Patients who consent to withdraw from eculizumab will receive their last dose of eculizumab during visit 1 (classed as day-14 prior to withdrawal).

## Visit details and assessments

### Baseline assessments and data collection for withdrawal cohort (visit 2, day 0±2 days)

Study day 0 will be the day that the participants would usually receive their next dose of eculizumab, based on standard dosing schedules (±2 days). The eculizumab will not be administered; however, meningococcal prophylaxis will be continued for a further 2 weeks after day 0.

At day 0 of the study (visit 2), participants will undergo the following assessments:

Vital signs (temperature, pulse and blood pressure), height and weight, renal function (creatinine and eGFR), urinalysis and urine protein/creatinine ratio, haemolysis markers including platelet count, haemoglobin, LDH, electrolyte profile, liver function (bilirubin, Alanine Aminotransferase (ALT)/Aspartate Aminotransferase (AST), Alkaline Phosphatase (ALP), Lactate Dehydrogenase (LDH), serum calcium, phosphate, albumin and total protein), haptoglobin (if available) and blood film, concomitant medication review, health-related quality of life questionnaires (EQ-5D-5L and SF-36) and healthcare utilisation questionnaire. A biomarker and complement activation sample to identify predictors of relapse (for example, soluble C5b-9) is also taken and stored at site before transfer to the Newcastle University.

### Study visit assessments and data collection for withdrawal cohort (visits 3–34)

Participants will be assessed regularly for evidence of disease relapse for the 2-year duration of the study. The participants will attend a total of 32 safety monitoring visits over the 2-year withdrawal follow-up period.

Trial participants will be reviewed at the trial site weekly (±2 days) for the first month, then alternate weeks (±2 days) until month 6, then monthly (±7 days) thereafter until the end of the trial period (month 24). At each study visit, the participants will undergo the monitoring assessments as detailed in figure 1.

Paediatric participants must have their weight recorded at every visit for calculation of eGFR. At the end of the trial, the level of safety monitoring for those patients who remain off treatment and disease free will be decided by their local clinical care team in discussion with the National aHUS Service.

Due to the COVID-19 pandemic, participants may be unable to attend their scheduled follow-up visits or may be attending a local hospital to have safety blood tests taken. If the participants are unable to attend site due to self-isolation or underlying health issues, where possible, a remote, follow-up call will be carried out by a member of the local research team. Participants will be asked to report changes to their concomitant medications, any AEs experienced since their previous follow-up and the results of their home urinalysis tests.

### Health economics assessments

Participants, or their parent/legal guardian, in both withdrawal and health economics cohorts will complete

**Table 1** Home urinalysis result thresholds

| Baseline haematuria result | Haematuria result threshold (not related to menstruation) |
|---|---|
| Neg/trace | ++ on any occasion OR + on any two occasions 24 hours apart |
| + | +++ on any occasion OR ++ on any two occasions 24 hours apart |
| ++ | +++ on any occasion |

the EQ-5D-5L (proxy version if patient <12 years), SF-36 (parent/legal guardian completes if patient <14 years) and a healthcare utilisation questionnaire at eight time points. A time and travel questionnaire is completed at one time point, as detailed in figures 1 and 2.

### Self-monitored urinalysis

Withdrawal participants, or their parent/legal guardian, will be trained to perform and understand the results of home urinalysis. Urinalysis, for the presence of haematuria or haemoglobinuria as an indicator of intravascular haemolysis and therefore disease activity, will then be performed daily by the participant or parent/legal guardian for the first month and then three times per week for the remainder of the study period. The results will be recorded in a participant diary and will be reviewed at each study visit. Participants or their parent/legal guardian will report any significant change in urinalysis, not related to menstruation, using their own baseline result to guide them in relation to the thresholds as detailed in table 1.

If the threshold criteria are met, participants or their parent/legal guardian will contact their treatment site immediately to arrange an unscheduled visit to assess disease activity as outlined in figure 1.

### Trial withdrawal

Participants will have the right to withdraw from the study at any time for any reason, and without giving a reason. The investigator will also have the right to withdraw participants from the study if she/he judges this to be in their best interest. Those participants who have been withdrawn from treatment can request to restart treatment, even if they have not relapsed. Data and blood samples provided by the participant up until the point of withdrawal will be included in analysis, unless they specifically request to have this removed. Participants who withdraw from the trial will not be replaced.

### Change in health status

Participants will be advised to report any significant change in health status to the responsible site or local healthcare provider. Participants will be provided with a participant identification card to present to attending medical staff with details of the study, tests required and study centre and National aHUS Service contact details (online supplemental file 2). Sites will notify the participants' general practitioner of their involvement in the

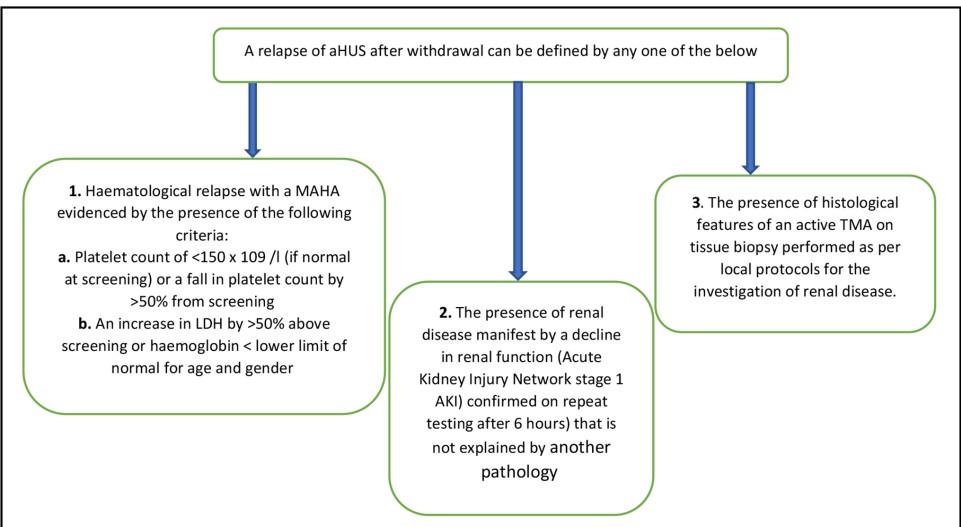

**Figure 3** Criteria for diagnosis of aHUS relapse. aHUS, atypical haemolytic uraemic syndrome; LDH, lactate dehydrogenase; MAHA, microangiopathic haemolytic anaemia; TMA, thrombotic microangiopathy.

study and inform them of the required action to be taken in the case of suspected relapse. Criteria for a diagnosis of aHUS relapse are shown in figure 3.

If there is a clinical suspicion of disease activity, formal assessment will occur as outlined in figure 1, unscheduled visit column.

Any other AEs that could represent a relapse will be discussed with the investigators and/or the National aHUS Service. A decision to restart will be made according to current service procedures.

### Relapse management

When a relapse is diagnosed, participants will restart eculizumab treatment within 24 hours of presentation provided there is no evidence of an active infection that would be a contraindication to treatment at the recommended dose of 900 mg weekly for the first 4 weeks then 1200 mg every 2 weeks thereafter (or age-adjusted dose and regime). TMA activity will be monitored (platelet count, LDH) as recommended by attending clinician until haematological remission is achieved as defined in figure 4.

Participants who relapse and require reintroduction of eculizumab treatment will remain on treatment in study under follow-up for the full 2 years of the study. Home urinalysis will not be required after reintroduction of eculizumab treatment.

Participants will consent to travel to only those countries where eculizumab is available. If a participant relapses while they are travelling outside of the country, the National aHUS Service will make arrangements with the destination country to access and fund eculizumab if required, with arrangements from the commissioning authority.

### Embedded qualitative study

In-depth one-to-one telephone interviews will be conducted following a topic guide developed with the input of the research team, including patient and public involvement (PPI). The intention is to keep interviews very broad to ensure we capture the full experience of interviewees.

Up to 30 patients who withdraw from eculizumab and up to 20 patients who decline to withdraw will be

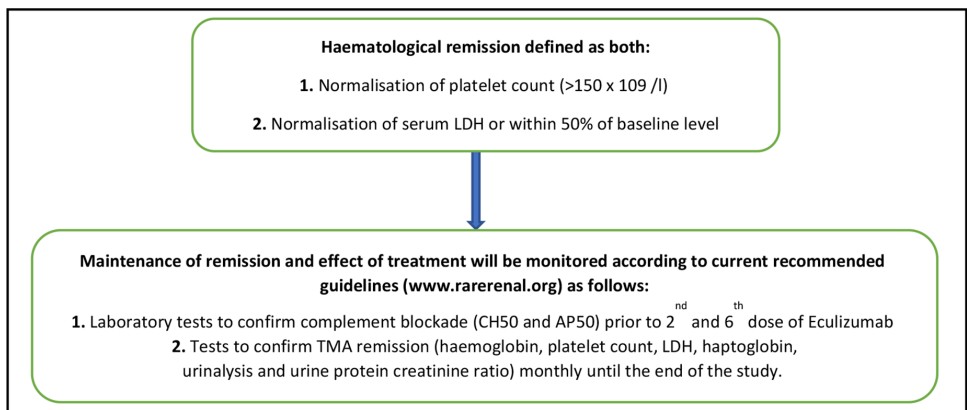

**Figure 4** Haematological remission definition and maintenance. LDH, lactate dehydrogenase; TMA, thrombotic microangiopathy.

approached to participate. Up to 20 patients who withdraw will be reinterviewed at the end of the withdrawal period (24 months later) to explore their views of the monitoring protocol. This group will be asked at the first interview if they agree to be contacted again towards the end of the study for a follow-up interview. Where possible, any patients who relapse and go back onto treatment will also be interviewed. Consent will be recorded at the time of the interview. Interviews will be digitally recorded with the permission of the interviewee, transcribed verbatim and anonymised.

### Safety reporting

All AEs occurring from the point of withdrawal (day 0) to end of study participation will be recorded. SAEs occurring from the point of withdrawal (day 0) must be reported to the Newcastle Clinical Trials Unit (NCTU) within 24 hours of the site becoming aware of the event. Serious adverse reactions (SARs) and suspected unexpected SARs are reportable only for those participants who have eculizumab treatment reinitiated during their participation in the trial. The assessment of expectedness will be performed by the principal investigator at site against the approved Reference Safety Information for the trial (Section 4.8 of the Soliris SmPC).

### Statistical analysis
#### Sample size

A maximum of 30 patients will be recruited to the withdrawal component; this is judged to be reasonable given the rare nature of the disease. The specifics of this sample size are intrinsically linked to the Bayes factor single-arm binary design employed to analyse the primary outcome measure. There is no allowance for loss to follow-up as this patient group is already subject to a high degree of clinical follow-up and death is defined as one of the serious events under consideration. Twenty patients will be recruited to the non-withdrawal arm as a comparator group for health economics analysis only.

#### Analysis

This is a single-arm, open-label trial with the primary endpoint being a binary response (the presence/absence of a primary outcome event within the follow-up period). We will compare the rate of TMA-related SAEs (primary outcomes) following the withdrawal of medication with that of treatment-related SAEs, expected under standard care.

The Bayes factor single-arm binary model[19] will be used to monitor the trial. Based on historical data, the event rate (treatment-related SAEs in 100 patients over a 2-year period) for the standard of care is 0.06, and we expect that withdrawal of the treatment would give a rate of 0.12. This choice of rate has been informed in discussion with patients. Using this Bayesian hypothesis test-based design, we assume the rate is 0.06 under the null, and 0.12 under the alternative hypothesis.

We assume that the sample distribution of number of responses follows a binomial distribution and use an inverse moment prior for response under the alternative hypothesis.

#### Stopping rules

A minimum of five patients will be enrolled before applying the stopping rules, and the cohort size for monitoring is five patients. The Data Monitoring Committee (DMC) can request earlier review if AEs are reported before this point.

We implement two stopping rules:
1. We will stop the trial for superiority (there being fewer TMA-related SAEs on the intervention than would be expected under standard of care) if the posterior probability of the alternative hypothesis is less than 0.05, that is, $\Pr(H1|Data) < 0.05$.
2. We will stop the trial for inferiority if the posterior probability of the alternative hypothesis is greater than 0.80, that is, $\Pr(H1|Data) > 0.80$.

#### Operating characteristics and stopping boundaries

The operating characteristics (table 2) and stopping boundaries (table 3) were produced using the M D Anderson Cancer Center Department of Biostatistics software BayesFactorBinary, V.1.0.[20]

If the true rate is 0.06 (scenario 1, null hypothesis), the trial will stop with probabilities of 0.096 and 0 in favour of the alternative and null hypotheses, respectively. The average number of patients (10%, 90% percentiles) is 28.44 (30, 30). If the true rate is 0.12 (scenario 2, alternative hypothesis), the trial will stop with probabilities of 0.443 and 0 in favour of the alternative and null

**Table 2** Operating characteristics

| Scenario | True rate of treatment-related serious adverse events | Probability of stopping for inferiority | Probability of stopping for superiority | Average number of patients treated (percentiles: 10%, 25%, 50%, 75%, 90%) |
|---|---|---|---|---|
| 1 | 0.06 | 0.096 | 0 | 28.44 (30, 30, 30, 30, 30) |
| 2 | 0.12 | 0.443 | 0 | 23.48 (5, 15, 30, 30, 30) |
| 3 | 0.18 | 0.753 | 0 | 17.76 (5, 10, 15, 30, 30) |
| 4 | 0.24 | 0.928 | 0 | 13.72 (5, 5, 15, 15, 30) |
| 5 | 0.30 | 0.982 | 0 | 10.52 (5, 5, 10, 15, 20) |

**Table 3** Stopping Eculizumab Treatment Safely in aHUS trial stopping boundaries

| Number of patients (in complete cohorts of 5) | Stop the trial for superiority if there are these many TMA events (inclusive) | Continue the trial if there are these many TMA events (inclusive) | Stop the trial for inferiority if there are these many TMA events (inclusive) |
|---|---|---|---|
| 5 | Never stop for superiority with these many patients | 0–1 | 2–5 |
| 10 or 15 | Never stop for superiority with these many patients | 0–2 | 3–15 |
| 20 | Never stop for superiority with these many patients | 0–3 | 4–20 |
| 25 or 30 | Never stop for superiority with these many patients | 0–4 (the trial always stops at 30 patients, which is the maximum) | 5–30 |

hypotheses, respectively. The average number of patients (10%, 90%) is 23.48 (5, 30).

The study will stop for inferiority with two TMA-related SAEs in the first cohort of five participants. Subsequently, the study would stop if three or more TMA-related SAEs are observed in the first 15 participants, four or more in the first 20 participants, and five or more in the whole study population. We are well placed to respond to any negative safety signal.

The 1000 repetitions were used in the software simulation. Calculations with different numbers of repetitions resulted in unchanged stopping boundaries with only marginal changes to the operating characteristics.

There may be differing risk of relapse according to disease aetiology. However, the available numbers do not allow for risk strata to be monitored separately. The DMC will consider this within their remit.

In addition to this ongoing analysis, at the end of the study, data will also be reported descriptively, together with the number of patients recruited. Descriptive statistics reported will be selected as appropriate to the specific outcome measure. For proportion outcomes, the number of patients recording the event will also be reported.

Due to the sample size, no comparative statistical methods will be applied. There will be no imputation of missing data and a complete case analysis will be undertaken.

### Subgroup analyses
Except for the analysis of the primary outcome on an ongoing basis, the analyses described above may be reported separately for different genetic groups or risk strata.

### Health economics analysis
#### Within-trial assessments of costs and outcomes
Costs and health outcomes (measured in terms of resource use of primary and secondary healthcare NHS and QALYs) associated with eculizumab withdrawal (30 participants), compared with standard care (20 participants), will be assessed over the 24-month follow-up period. Information on costs and health outcomes will be recorded for each individual involved in both treatment groups. Data derived from the within-trial analysis will be assessed to understand the key determinants of differences in costs and outcomes between the two patient groups. Data will then be used to parameterise

the lifetime economic model (combined with data from the literature).

### Assessment of cost-effectiveness
An economic decision model will be developed to assess the cost-effectiveness of the alternative treatment options under evaluation. Costs and health consequences, measured in terms of QALYs, associated with eculizumab withdrawal, and a policy of monitoring following withdrawal, and standard care, beyond the 2-year time frame of the trial will be captured. We propose conducting a cost–utility analysis, with results presented in terms of incremental cost per QALY gained.

### Qualitative analysis
We will take an inductive approach to data collection and analysis. This means there is no a priori theory; themes, concepts and theories will be elicited from the interview data when it is analysed and drawing upon relevant literature, PPI and experts in the study team. Data will be analysed thematically using a constant comparative method. This entails a process of familiarisation with the data and then the development of a thematic framework. A small number of transcripts will be coded, and the framework amended accordingly. A second level analysis will be conducted using a constant comparative method. This involves a process of comparing and contrasting themes elicited from the data, within and across interviews.[21] NVivo will be used as a data management tool.

### Trial management and monitoring
This trial is sponsored by the Newcastle upon Tyne Hospitals NHS Foundation Trust. The trial will be coordinated by a TMG that will include those individuals responsible for the day-to-day management of the trial.

A Trial Steering Committee (TSC) made up of independent clinical and lay members will provide overall supervision of the trial. A DMC composed of independent clinicians and statisticians will undertake independent review and monitor efficacy and safety endpoints. The trial was prospectively registered on the International Standard Registered Clinical/soCial sTudy Number Registry and the European Union Drug Regulating Authorities Clinical Trials Database (online supplemental file 3).

The NCTU will be responsible for communicating protocol amendments to participating sites and carrying out central, remote and on-site monitoring.

## Confidentiality and data handling

Personal data will be regarded as strictly confidential. To preserve anonymity, a unique participant ID will be assigned to each participant at consent. Only the clinical team at the participating sites will have access to key data which link study identifiers to individual datasets. All study records and investigator site files will be kept at site in a locked filing cabinet with restricted access.

Written consent will be sought from participants or legal guardians, if patient is under the age of 16 years, to allow access to their hospital records.

Data are recorded by authorised staff and stored in a secure web-based electronic data capture system (MACRO) designed and maintained by NCTU hosted on secure servers at Rackspace within the UK. Analysis of the data will be undertaken by the Newcastle University trial statisticians. Anonymised data from this trial may be available to the scientific community subject to regulatory and ethics approval. Requests for data should be directed to the corresponding author. All study data will be archived for 5 years.

## Patient and public involvement

A PPI representative sits on the TMG, was involved in protocol and study document development, and is involved in ongoing trial management discussions. We also have a patient with aHUS as an independent member of the TSC.

## Ethics and dissemination

A favourable ethical opinion and approval was obtained from the North East-Tyne & Wear South Research Ethics Committee in April 2018. Written informed consent will be obtained from all participants prior to their involvement in the trial. The results of the study will be submitted to peer-reviewed journals, presented at conferences and on the trial website.

## DISCUSSION

This study will determine whether it is safe to withdraw eculizumab using a trial methodology designed to detect an excess of adverse outcomes following withdrawal (primary endpoint). The study will also estimate the proportion of patients with aHUS that can be maintained off eculizumab and test a system for surveillance to identify relapse early (secondary endpoints). This will allow a cost–utility analysis to be conducted, exploring the impact of treatment withdrawal.[22] This carefully monitored patient group will allow us to determine how early subclinical relapse can be detected using standard biochemical and haematological measurements and novel biomarkers of complement activation or tissue injury. An embedded qualitative study of patients, both those who withdraw and decide not to withdraw, will explore attitudes towards treatment and its withdrawal.

## Trial status

This manuscript is based on trial protocol version 7.0 dated 14 January 2021. The first patient was recruited in November 2018, recruitment ended on 31 January 2022 and planned last patient visit is November 2023.

**Author affiliations**
[1]Newcastle Clinical Trials Unit, Newcastle University, Newcastle upon Tyne, UK
[2]National Renal Complement Therapeutics Centre, Newcastle University, Newcastle upon Tyne, UK
[3]Population Health Sciences Institute, Newcastle University, Newcastle upon Tyne, UK
[4]Great North Children's Hospital, Newcastle upon Tyne Hospitals NHS Foundation Trust, Newcastle upon Tyne, UK
[5]Translational and Clinical Research Institute, Newcastle University, Newcastle upon Tyne, UK
[6]Health Economics, Optimax Access, Newcastle upon Tyne, UK
[7]aHUS Alliance Global Action, Newcastle upon Tyne, UK

**Acknowledgements** The authors would like to thank Dr Colin Muirhead who was responsible for the statistical proposal in the initial grant application. The authors would also like to thank all the principal investigators for their contribution and support, the research nurses, the members of the TSC and DMC, and all the patients involved in the SETS aHUS trial.

**Contributors** NS is the chief investigator and senior author of the SETS aHUS design and has led the grant acquisition and protocol development. DK, SJ, EKSW, LW, JL, TJC and YO are co-applicants of the grant and contributed to the design of the trial and protocol development. SC and SD are part of the trial management team and contributed to protocol development. EM is a health economist and contributed to protocol development. AB is a research statistician and contributed to protocol development. CW is a data manager and contributed to protocol development. MM and VB are part of the National Renal Complement Therapeutics Centre and have played key roles in the running of the SETS aHUS trial. This paper was drafted from the current approved version of the protocol by the corresponding author SD. All contributing authors commented and amended drafts of the paper. All contributing authors read and approved the final manuscript.

**Funding** The SETS aHUS trial is funded by the National Institute for Health Research (NIHR) Health Technology Assessment (HTA) Programme (project number 15/130/94).

**Disclaimer** The views expressed are those of the authors and not necessarily those of the NIHR or the Department of Health and Social Care.

**Competing interests** SD has received honoraria for sitting on advisory boards for Alexion and Novartis. DK is a director of and scientific advisor to Gyroscope Therapeutics. DK received advisory board payments from Idorsia, Novartis, ChemoCentryx, Alexion, Apellis, Biomarin and Sarepta. DK's spouse works for GSK. MM has received honoraria for educational talks and honorarium for national lead of aHUS registry, both from Alexion and travel expenses from Alexion. EKSW has received honoraria for lectures and/or advisory boards for Alexion Pharmaceutical, Biocryst and Novartis. LW has received expenses, honoraria and fees for advisory board participation and talks from Alexion and Roche. NS has given lectures or sat on advisory boards for Alexion Pharmaceutical, Roche, AstraZeneca and Novartis; no personal honoraria, all payments made to the department.

**Patient and public involvement** Patients and/or the public were involved in the design, or conduct, or reporting, or dissemination plans of this research. Refer to the Methods section for further details.

**Patient consent for publication** Not required.

**Provenance and peer review** Not commissioned; externally peer reviewed.

**ORCID iDs**
Sarah Dunn http://orcid.org/0000-0002-1901-3931
Michal Malina http://orcid.org/0000-0002-9434-1521

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
