## [Reviewer comments · BMJ Open]

ARTICLE DETAILS

TITLE (PROVISIONAL)	Protocol for a multicentre, open label, prospective, single arm study of the safety and impact of eculizumab withdrawal in patients with atypical haemolytic uraemic syndrome
AUTHORS	Dunn, Sarah; Brocklebank, Victoria; Bryant, Andrew; Carnell, Sonya; Chadwick, Thomas; Johnson, Sally; Kavanagh, David; Lecouturier, Jan; Malina, Michal; Moloney, Eoin; Oluboyede, Yemi; Weetman, Christopher; Wong, Edwin; Woodward, Len; Sheerin, Neil

VERSION 1 – REVIEW

REVIEWER	Krishnappa, Vinod Southeastern Health, Internal Medicine
REVIEW RETURNED	29-Jul-2021

GENERAL COMMENTS	aHUS is a rare, yet deadly disease with high mortality and morbidity rates. High index of clinical suspicion is need for its early diagnosis and requires expert management. The pathophysiology was poorly understood in the past but in the last decade and a half, there has been extensive research, which lead to the development of life saving drug eculizumab. However, the major limitation is its cost. Further, duration of eculizumab treatment is poorly defined and majority of clinical trials tested this drug for 6-24 months. The authors of this protocol propose to study safety and impact of eculizumab withdrawal, and cost-effectiveness of continuing treatment for long-term in a structured manner. They also propose to reintroduce eculizumab in patients who relapse after withdrawal by closely following them for features of relapse. The authors described robust study methodology as well. This is an important and very needed study to determine the duration of treatment with this very expensive drug, eculizumab, and also to guide reintroduction of this drug to patients who relapse. Few minor comments are below for authors Page 4, line 12, "...innate immune system and in health is tightly regulated.." correct "health" to "healthy" On page 4 , lines 44 & 45 are ambiguous "In the first year after presentation, 25% of children and 29% of adults will experience a relapse. 82% of relapses in adults, and 57% of relapses in children, occur in the first year after disease onset". This needs more clarity. There are two different percentages of relapses reported in children and adults in the first year following presentation. Which is correct? On page 5, lines 44-46, secondary objective (No. 9), authors say "To model the lifetime costs and outcomes associated with Eculizumab
--

	withdrawal, and a policy of protocolised monitoring following withdrawal (and treatment re-introduction if necessary), compared with standard care, beyond the two-year timeframe of the trial.” How will this study assess lifetime costs and outcomes when the study period is 24 months? On page 6, under eligibility for patient enrollment into the study, “On Eculizumab treatment for at least 6 months”. Given majority of aHUS patients relapse in the first year of presentation, would it be prudent to enroll patients who have completed at least 12 months of eculizumab treatment? On page 7, lines 15-18, under intervention “Patients who consent to withdraw from Eculizumab will receive their last dose of Eculizumab at this visit (day -14).” This is little confusing. Is this visit number 1? How is this day-14? Next under Visit Details and Assessments, it says “Study day 0 will be the day that the participants would usually receive their next dose....” How is this day 0 if the previous visit is day-14? Authors need to clarify this.
--	---

REVIEWER	Alabdulqader, Muneera King Faisal University
REVIEW RETURNED	08-Aug-2021

GENERAL COMMENTS	Thank you for allowing me to review the manuscript. The study is similar to previously published studies. The findings are expected to be similar. After reading the submitted protocol, I have two main questions. 1- do all the participants have the genetic study done before enrollment? Since some genetic variants (especially MCP and CFH genes mutations) are at higher risk of relapse after Eculizumab discontinuation, Was this fact considered in enrolling the patients? 2. What are the complement activation samples consist of? Fakhouri et al. Found that “an elevated sC5b-9 level at eculizumab discontinuation was independently associated with the risk of aHUS relapse in multivariable analysis.”. It will be very interesting if your research will have similar findings.
---

REVIEWER	Roberts, Matthew Monash University Eastern Health Clinical School
REVIEW RETURNED	08-Dec-2021

GENERAL COMMENTS	This study seeks to address an important question with significant patient and health economic consequences. I have a number of questions that I don't feel are sufficiently explained for the reader to fully understand the study: Major queries: 1. The authors describe the Primary outcome as Serious Adverse Events in terms of a specific loss of kidney function. The rationale for this could be described in more depth: Who will decide if a reduction in eGFR is not attributable to another cause, and how? Why was a 20% change chosen? 2. The primary outcome and serious adverse events terminology is used interchangeably. I agree that a loss of kidney function is an important and serious adverse event. There are other serious adverse events – death, admissions for infection, consequences of
--

	anaemia of thrombocytopenia. Will relapse requiring re-treatment with eculizumab be considered an adverse event, or only of the change in GFR criteria are met? 3. It is ambiguous to me whether the analysis is about the primary delta GFR outcome, or all serious adverse events. The same is true for the stopping rules. 4. Is this a non-inferiority trial? The probability of stopping for superiority is zero under all scenarios in the Table. This is not surprising given the number of participants. 5. The event rate for the standard of care is stated as 0.06, but what this means should be explicitly stated, with units/time. I interpret this as meaning 6% of patients per year will develop an irreversible 20% loss of eGFR not attributable to other things in the standard of care group receiving eculizumab. By definition, this event will take 3 months to develop after the first loss of GFR is noted. In aHUS, 3 months is enough time to lose a lot more kidney function than 20%! 6. Could the authors please explain the rationale for the use of urinalysis in more depth. Are there characteristic findings on urinalysis in TMA? Is urinalysis a good “screening test” for development of TMA? It is not a component of the criteria to diagnose a relapse (Figure 3), or measured at the unscheduled visit for suspected relapse (Table 1). Does Table 1 refer to blood, protein or another urinalysis parameter? There are approximately 174 home urinalysis measurements, which may generate some false positive results. Will these unscheduled visits be counted in the economic analysis? 7. Will recommencement of eculizumab be recommended to people who choose to withdraw? What advice will be given about ongoing monitoring if they stop the study early and decline to follow the protocol? 8. How can the authors be sure that participants will restart eculizumab within 24 hours of presentation, given that they will present to centres across the country and very possibly outside of usual business hours. What mechanisms are in place to ensure that this happens? 9. Do the authors anticipate any limitations to this study? Minor queries: 1. Does this study have Ethics Committee approval? A favourable opinion is not the same as approval with Ethics Committee reference. 2. Please clarify recruitment end date – 31 August 2021 according to Abstract page. 3. Eligibility: Wouldn't the fourth dot point be better expressed as $eGFR > 30 \text{ mL/min/1.73m}^2$? 4. Can people receiving eculizumab relapse? Is this data being collected? 5. Even if a destination country can access eculizumab, can the authors be sure that this can be accessed in a timely fashion? Will the NHS pay for this at the price the local health service charges? Will patients be given some sort of medical information/certificate to take with them to assist timely access to eculizumab? 6. What factors and assumptions will be used in the Health Economic analysis? Will resource use include the cost of monitoring (i.e. procedures in Figure 1)? Will patient time/time off work be factored in? 7. How often will the DMC meet and review the data? Will this be determined by time, or number of participants recruited? 8. The web link at the bottom of page 12 is a link to a document that could be referenced in the Reference List in my view. If it were the simulating software or an online program for the readers to explore
--	---

	and test the authors' numbers, then a link would be appropriate. Would be guided by the Editors about this.
--	---

REVIEWER	Yau , Kevin University of Toronto, Nephrology
REVIEW RETURNED	12-Dec-2021

GENERAL COMMENTS	This is a thoughtfully designed protocol for a multi-centre open label single-arm study examining the safety and impact of Eculizumab withdrawal in patients with aHUS. This study aims to address an important question given the substantial cost of lifetime treatment with Eculizumab and incorporates a Bayesian design due to the rare nature of the disease. I have only a few comments/questions for the authors:  1. The authors note that there is no allowance for loss to follow-up due to close monitoring of this patient population and given that death is an outcome in this study. However, despite close follow-up it is often difficult to obtain 100% follow-up. 2. It is noted that patients with aHUS will fulfill criteria determined by the National aHUS service. Could the authors include the exact criteria for diagnosis, and how other causes of thrombotic microangiopathy are excluded including the specifics of genetic testing that is performed? 3. Was there any consideration of excluding patients with CFH or MCP variants given that these patients are considered at higher risk of relapse (even if they are not transplant recipients)? 4. A recent study (doi: 10.1182/blood.2020009280) identified that the rate of relapse in a cohort of children/adults with aHUS was 13/55 (23%) with the majority of relapsing patients having variants in CHF or MCP. Could the authors clarify the definition of "serious events" being used for the stopping rules? With the current stopping rules the trial would be stopped for inferiority for example if there were 4 serious events in 20 patients according to the table presented. Might the current stopping rules risk ending the trial too early if the true rate of relapse reaches 23%, even if there is a genetic subset (i.e. patients without CFH or MCP pathogenic variants) who might be able safely discontinue Eculizumab? It may also be worth noting how your inclusion criteria or outcome assessments differ from the recently published study above.  5. What is tested in the biomarkers and complement activation sample? Does this include serum sC5b-9? 6. Have there been any other important modifications to the trial in light of the COVID-19 pandemic other than the study visits such as recruitment, trial pause, or other protocol modifications? If any, consider following this guideline: https://jamanetwork.com/journals/jama/fullarticle/2781397
--

VERSION 1 – AUTHOR RESPONSE

Reviewer: 1

Dr. Vinod Krishnappa, Southeastern Health

Comments to the Author:

aHUS is a rare, yet deadly disease with high mortality and morbidity rates. High index of clinical suspicion is need for its early diagnosis and requires expert management. The pathophysiology was poorly understood in the past but in the last decade and a half, there has been extensive research, which lead to the development of life saving drug eculizumab. However, the major limitation is its cost. Further, duration of eculizumab treatment is poorly defined and majority of clinical trials tested this drug for 6-24 months.

The authors of this protocol propose to study safety and impact of eculizumab withdrawal, and cost-effectiveness of continuing treatment for long-term in a structured manner. They also propose to reintroduce eculizumab in patients who relapse after withdrawal by closely following them for features of relapse. The authors described robust study methodology as well. This is an important and very needed study to determine the duration of treatment with this very expensive drug, eculizumab, and also to guide reintroduction of this drug to patients who relapse.

Few minor comments are below for authors

1. Page 4, line 12, "...innate immune system and in health is tightly regulated.." correct "health" to "healthy"

This sentence has now been changed to: The complement system is part of the innate immune system and in healthy individuals is tightly regulated to prevent excessive activation.

2. On page 4 , lines 44 & 45 are ambiguous "In the first year after presentation, 25% of children and 29% of adults will experience a relapse. 82% of relapses in adults, and 57% of relapses in children, occur in the first year after disease onset". This needs more clarity. There are two different percentages of relapses reported in children and adults in the first year following presentation. Which is correct?

This sentence has been amended to reduce any ambiguity and now reads:

From experience prior to the introduction of Eculizumab, the risk of relapse is greatest in the period immediately after first presentation with 82% of relapses in adults, and 57% of relapses in children occurring within the first year after disease onset.

3. On page 5, lines 44-46, secondary objective (No. 9), authors say "To model the lifetime costs and outcomes associated with Eculizumab withdrawal, and a policy of protocolised monitoring following withdrawal (and treatment re-introduction if necessary), compared with standard care, beyond the two-year timeframe of the trial." How will this study asses lifetime costs and outcomes when the study period is 24 months?

Beyond the two year trial period, different statistical models will be used to extrapolate study data for the life-time duration of the patients, following NICE DSU guidelines (1), and using data from the literature and expert opinion to validate the models and using sensitivity analysis to explore the uncertainty of the results. Cost and outcomes data will be presented in line with the latest NICE guidelines (2).

4. On page 6, under eligibility for patient enrollment into the study, "On Eculizumab treatment for at least 6 months". Given majority of aHUS patients relapse in the first year of presentation, would it be prudent to enroll patients who have completed at least 12 months of eculizumab treatment?

Thank you for this comment. Although, as pointed out by the reviewer, relapses are most likely to occur within the first year, the highest risk is in fact earlier after presentation. This is also true in relapses that occur after transplantation (3). It was therefore decided to set the minimum period of treatment at 6 months. As we are recruiting from a prevalent, treated patient population, some patients will be on treatment for longer than six months.

5. On page 7, lines 15-18, under intervention “Patients who consent to withdraw from Eculizumab will receive their last dose of Eculizumab at this visit (day -14).” This is little confusing. Is this visit number 1? How is this day-14? Next under Visit Details and Assessments, it says “Study day 0 will be the day that the participants would usually receive their next dose....” How is this day 0 if the previous visit is day-14? Authors need to clarify this.

The final eculizumab dose is administered during study visit 1. Participants will be due their next dose 14 days later during study visit 2. This dose is not given and therefore study visit 2 is classed as day 0 of the withdrawal protocol. On this basis study visit 1 occurs on day -14. The section has been clarified to reflect this:

Patients who consent to withdraw from Eculizumab will receive their last dose of Eculizumab during study visit 1 (classed as day -14 prior to withdrawal).

Reviewer: 2

Dr. Muneera Alabdulqader, King Faisal University Comments to the Author:

Thank you for allowing me to review the manuscript.

The study is similar to previously published studies. The findings are expected to be similar.

After reading the submitted protocol, I have two main questions.

1. do all the participants have the genetic study done before enrollment? Since some genetic variants (especially MCP and CFH genes mutations) are at higher risk of relapse after Eculizumab discontinuation, Was this fact considered in enrolling the patienta?

All potential participants in the study will have had genetic testing for clinically significant variants in complement regulatory proteins (including C3 and CFB) and for anti-CFH autoantibodies as per standard of care in the UK. However, all patients who are on eculizumab for the treatment of aHUS will be eligible to participate in the trial irrespective of whether an abnormality is found as they will have met the criteria for treatment. The only exception to this is with transplant recipients who are excluded if they have a pathogenic variant in CFH, C3 or CFB.

2. What are the complement activation samples consist of? Fakhouri et al. Found that “an elevated sC5b-9 level at eculizumab discontinuation was independently associated with the risk of aHUS relapse in multivariable analysis.”. It will be very interesting if your research will have similar findings.

Soluble C5b-9 will be measured, but other markers of complement activation will also be considered including Bb, Ba and C3a. These are exploratory and will be analysed after the study is complete. This has been clarified in the text (page 7).

Reviewer: 3

Dr. Matthew Roberts, Monash University Eastern Health Clinical School Comments to the Author:

This study seeks to address an important question with significant patient and health economic consequences. I have a number of questions that I don't feel are sufficiently explained for the reader

to fully understand the study:

Major queries:

1. The authors describe the Primary outcome as Serious Adverse Events in terms of a specific loss of kidney function. The rationale for this could be described in more depth: Who will decide if a reduction in eGFR is not attributable to another cause, and how? Why was a 20% change chosen?

We apologise that we have omitted some of the detail in the primary outcome section (page 5). TMA related SAEs should also include AKI that requires renal replacement therapy and any non-renal manifestation of a TMA that requires hospitalisation. A participant will reach a primary endpoint if one of these three criteria are met. A SAE not related to a TMA or relapse in TMA that does not fulfil any of these criteria will not be classed as an end point.

If a relapse is suspected then the patient should be referred to the national aHUS service, based in Newcastle. This provides a 24 hour, 7 day a week service and will decide, along with the local clinical team, as to whether the presentation represents a relapse and therefore whether eculizumab should be restarted. This will be the process by which additional information is requested to ensure the information available is sufficient to determine whether a relapse has occurred.

In addition, there will also be real time monitoring of adverse events using information collected from study sites via the trial database. The data manager will generate regular (every 2 weeks) reports including renal function, platelet count and LDH, which is submitted to the CI for review.

The DMC will have final say as to whether a relapse has occurred (based on recommendation from the expert centre) and whether any of three criteria listed as TMA related SAEs have occurred.

A 20% loss of function was chosen based on discussion with the study team. It was felt that this loss of function was clinically significant and that a persistent fall in eGFR was likely to represent irreversible renal injury. Other criteria that have been suggested, for example by KDIGO, define complete recovery as an eGFR $>60\text{mls/mi}/1.73\text{m}^2$ which would not be appropriate for this population.

2. The primary outcome and serious adverse events terminology is used interchangeably. I agree that a loss of kidney function is an important and serious adverse event. There are other serious adverse events – death, admissions for infection, consequences of anaemia or thrombocytopenia. Will relapse requiring re-treatment with eculizumab be considered an adverse event, or only if the change in GFR criteria are met?

As in the answer to the first point, the submitted manuscript did not include a full description of the primary outcome measure. The primary outcome is not limited to a decline in renal function but includes other TMA manifestations that require hospitalisation, renal replacement therapy or death.

Unlike other studies we do not include relapse as a primary outcome unless it is associated with one of the TMA related SAEs listed in the manuscript. We expect that patients will relapse after withdrawal of eculizumab and describing the rate of relapse is a secondary objective of the study.

We have identified places in the manuscript where serious adverse event and primary outcome are used interchangeably, and either corrected or provided clarification (pages 9 and 11).

3. It is ambiguous to me whether the analysis is about the primary delta GFR outcome, or all serious adverse events. The same is true for the stopping rules.

Again, we apologise for this ambiguity which is due to the description of the primary outcome measure being incomplete. The broader definition of the primary outcome (TMA related SAE) is now

included in the manuscript. This is the patient outcome that is applied to the stopping rule.

4. Is this a non-inferiority trial? The probability of stopping for superiority is zero under all scenarios in the Table. This is not surprising given the number of participants.

This is not strictly a non-inferiority design – with a larger sample size we may have been able to stop for superiority. However, in practical terms, for the numbers we are able to recruit, we are only able to stop for inferiority.

5. The event rate for the standard of care is stated as 0.06, but what this means should be explicitly stated, with units/time. I interpret this as meaning 6% of patients per year will develop an irreversible 20% loss of eGFR not attributable to other things in the standard of care group receiving eculizumab. By definition, this event will take 3 months to develop after the first loss of GFR is noted. In aHUS, 3 months is enough time to lose a lot more kidney function than 20%!

The event rate in the standard of care population was obtained by review of the first 100 patients treated with eculizumab under care of the national aHUS service in the UK. In the first two years of their treatment 6 serious events were identified that were either definitely or probably associated with eculizumab treatment (by expert review). The most frequently identified event was meningococcal infection. This rate (0.06 in a 2 year follow period) was set as the treatment-associated event rate in the standard of care population. The type of serious event that occurred in the standard of care population will therefore be different from the events expected after withdrawal. We would not expect patients on eculizumab to develop a thrombotic microangiopathy (see below), therefore, the primary outcome (TMA related SAE) will only be seen after withdrawal.

This has been clarified in the manuscript (page 9).

We agree that renal function can deteriorate rapidly in aHUS and AKI requiring renal replacement therapy would be a primary outcome. A lesser degree of AKI with recovery of function to within 20% of baseline would meet the threshold to be classed a primary outcome. However, if there was incomplete recovery with loss of >20% of function a primary outcome would be met.

6. Could the authors please explain the rationale for the use of urinalysis in more depth. Are there characteristic findings on urinalysis in TMA? Is urinalysis a good “screening test” for development of TMA? It is not a component of the criteria to diagnose a relapse (Figure 3), or measured at the unscheduled visit for suspected relapse (Table 1). Does Table 1 refer to blood, protein or another urinalysis parameter? There are approximately 174 home urinalysis measurements, which may generate some false positive results. Will these unscheduled visits be counted in the economic analysis?

There has been a report suggesting that the development of haemoglobinuria is an early indicator of disease activity, indicating the presence of intravascular haemolysis (4). This is from case series and has not been assessed as part of a formal clinical trial. Therefore, whether or not urinalysis for haematuria/haemoglobinuria is good ‘screening test’ for TMA relapse is not known. It will be assessed as part of the proposed protocol. The rationale for urinalysis has been explained on page 8 of the manuscript.

We agree that haematuria/haemoglobinuria is not part of the standard criteria to diagnose a relapse of aHUS. Home testing will potentially allow patients to monitor for the presence of relapse and if a change in the level of haematuria/haemoglobinuria occurs it will trigger an unscheduled visit. Table 1 has been amended to clarify that urinalysis is performed to look for haematuria or haemoglobinuria. At the unscheduled visit tests will be performed to assess for relapse.

The health economics analysis will factor in both scheduled and unscheduled visits – using data collected from the Health Care Utilisation (HCU) questionnaire and electronic Case Report Forms (eCRFs).

7. Will recommencement of eculizumab be recommended to people who choose to withdraw? What advice will be given about ongoing monitoring if they stop the study early and decline to follow the protocol?

If patients withdraw from the study, they will be given the option to restart eculizumab. This decision will be made following discussions between the patient, the local responsible clinical team, and the national aHUS service.

8. How can the authors be sure that participants will restart eculizimab within 24 hours of presentation, given that they will present to centres across the country and very possibly outside of usual business hours. What mechanisms are in place to ensure that this happens?

Relapses will be managed in the same way as other patients presenting with aHUS. The patient will be referred to the national aHUS service. If appropriate for the patient to restart treatment, the on-call consultant will work with the local team, the manufacturers and other hospitals to ensure that eculizumab is available.

9. Do the authors anticipate any limitations to this study?

Some limitations of the study are listed as part of the abstract on page 3 of the manuscript. A major limitation is the small number of patients available for a study in such a rare disease. It is therefore not possible to perform a randomised control trial of safety. This is acknowledged on page 3. However, eculizumab was licenced for the treatment of aHUS after similar single arm studies. We therefore feel that this is an appropriate study design.

Minor queries:

1. Does this study have Ethics Committee approval? A favourable opinion is not the same as approval with Ethics Committee reference.

We received a favourable opinion from the Research Ethics Committee followed by HRA approval. These are standard approvals for the UK. The REC favourable opinion letter and HRA approval letter have both been submitted for information.

2. Please clarify recruitment end date – 31 August 2021 according to Abstract page.

At the time of submission of the manuscript, the recruitment end date was 31 August 2021, the recruitment period was further extended to 31 January 2022 and so the manuscript has been updated to reflect this.

3. Eligibility: Wouldn't the fourth dot point be better expressed as $eGFR > 30 \text{ mL/min/1.73m}^2$?

We agree that it should be clarified that an eGFR of $>30 \text{ ml/min/1.73m}^2$ is required to participate in the study. This has been added to the text.

4. Can people receiving eculizimab relapse? Is this data being collected?

If patients receiving eculizumab have a relapse with evidence of a thrombotic microangiopathy this will indicate either an insufficient dose of eculizumab or an alternative diagnosis. We would not expect

patients with a primary complement-mediated form of aHUS to relapse whilst adequately dosed with eculizumab. Clinical data is only collected in the withdrawal arm of the study. However, there is a shared care agreement for all patients on eculizumab in England and Scotland with monitoring of patients through the national specialised clinical service. The CI would therefore be aware of any patient who relapses whilst on treatment, irrespective of whether they are participating in this trial.

5. Even if a destination country can access eculizumab, can the authors be sure that this can be accessed in a timely fashion? Will the NHS pay for this at the price the local health service charges? Will patients be given some sort of medical information/certificate to take with them to assist timely access to eculizumab?

The National aHUS service will work with local clinical services and Alexion Pharmaceuticals to allow initiation of treatment in a short a time as possible. The NHS has agreed to pay the costs of eculizumab if the patient is abroad at the time of a relapse occurring as part of this trial.

Patients are provided with a card containing the following information:

The holder of this card has a rare disease known as atypical Haemolytic Uraemic Syndrome and is taking part in a clinical trial to assess the safe withdrawal of Eculizumab treatment. Because of this, the holder of this card may suffer a relapse.

If the holder presents unwell, however minor the illness, please evaluate immediately and obtain the following Laboratory investigations as their Eculizumab treatment and prophylactic antibiotics may need to be re-started as soon as possible and within 24hrs:

- U&E • FBC • LDH

If the results are abnormal, immediately contact the local medical team and refer to the UK National aHUS Service website for advice.

The contact number for the UK national aHUS service and local team is stated on the card. This card is included with the manuscript as Supplement 2 and referred to in the body of the text (page 8).

6. What factors and assumptions will be used in the Health Economic analysis? Will resource use include the cost of monitoring (i.e. procedures in Figure 1)? Will patient time/time off work be factored in?

The health economics study will follow good practice guidance, the factors and assumptions that will be considered will be informed by the latest NICE health technology assessment guideline (5). The resource use relating to monitoring will be included in the analysis, utilising micro-costing methods. The patient time/time off work will be factored in by using data collected in the Health Care Utilisation (HCU) and Time and Travel (T&T) questionnaires (implemented as per figure 2).

7. How often will the DMC meet and review the data? Will this be determined by time, or number of participants recruited?

The DMC meet at least annually as per the charter agreed and signed at the first meeting. Additional DMC meetings can be called to review data if required because of patients possibly meeting the criteria for primary endpoints.

8. The web link at the bottom of page 12 is a link to a document that could be referenced in the Reference List in my view. If it were the simulating software or an online program for the readers to explore and test the authors' numbers, then a link would be appropriate. Would be guided by the Editors about this.

We have moved this link to the reference section as suggested.

Reviewer: 4

Dr. Kevin Yau , University of Toronto

Comments to the Author:

This is a thoughtfully designed protocol for a multi-centre open label single-arm study examining the safety and impact of Eculizumab withdrawal in patients with aHUS. This study aims to address an important question given the substantial cost of lifetime treatment with Eculizumab and incorporates a Bayesian design due to the rare nature of the disease.

I have only a few comments/questions for the authors:

1. The authors note that there is no allowance for loss to follow-up due to close monitoring of this patient population and given that death is an outcome in this study. However, despite close follow-up it is often difficult to obtain 100% follow-up.

We agree that we cannot be certain of 100% follow up in this study. However, as stated in the protocol this patient group is under regular review as part of their standard of care and if patients withdraw from treatment there will be fewer contacts with healthcare professionals, therefore the burden of trial visits is minimised. If patients do withdraw it will indicate that the proposed strategy is not safe or effective and this will in itself provide useful information.

2. It is noted that patients with aHUS will fulfill criteria determined by the National aHUS service. Could the authors include the exact criteria for diagnosis, and how other causes of thrombotic microangiopathy are excluded including the specifics of genetic testing that is performed?

There is a NHS England and Scotland commissioned service that performs all of the diagnostics for a HUS and authorises treatment with eculizumab. The service has shared arrangements with local units to manage these patients and maintains a register of all patients on treatment. For authorisation of treatment patients must fulfil a series of inclusion and exclusion criteria. These include the exclusion of other causes of a thrombotic microangiopathy (STEC infection, TTP and secondary TMAs, etc). All patients have full genetic screening of pathogenic variants in CFH, C3, CD46, CFH, CFI and for non-complement causes of a TMA (DGKε, MMACHC). The presence of anti-FH antibodies is also assessed.

We feel that inclusion of these details is beyond the scope of this paper and have been published elsewhere (6). We have therefore included this reference in the manuscript (page 7).

3. Was there any consideration of excluding patients with CFH or MCP variants given that these patients are considered at higher risk of relapse (even if they are not transplant recipients)?

A decision was made to include patients with CFH and MCP variants despite their risk of relapse. The trial is to test the safety of withdrawal in patients including the potential to reintroduce eculizumab therapy in those patients who do relapse, including patients at risk of relapse is therefore critical for the trial. In addition, data suggests that although these patients are at greater risk of relapse not all patients will relapse, and a proportion will be stable off treatment.

4. A recent study (doi: 10.1182/blood.2020009280) identified that the rate of relapse in a cohort of children/adults with aHUS was 13/55 (23%) with the majority of relapsing patients having variants in CHF or MCP.

Could the authors clarify the definition of "serious events" being used for the stopping rules? With the current stopping rules the trial would be stopped for inferiority for example if there were 4 serious

events in 20 patients according to the table presented. Might the current stopping rules risk ending the trial too early if the true rate of relapse reaches 23%, even if there is a genetic subset (i.e. patients without CFH or MCP pathogenic variants) who might be able safely discontinue Eculizumab? It may also be worth noting how your inclusion criteria or outcome assessments differ from the recently published study above.

A relapse for which treatment is restarted and there is no irreversible loss of renal function, requirement for RRT or other tissue injury will not be classed as a primary endpoint. We expect a rate of relapse in the region of 25-30% as suggested by the reviewer, but we expect that rate of patients reaching a primary endpoint will be significantly lower.

As in response to reviewer 3 we apologise for not including the full definition of a primary outcome, which now includes AKI requiring renal replacement therapy and significant involvement of another organ. This is the main difference between this trial and the recently published trial from France, which used relapse as the primary endpoint. Relapse is expected in patients who withdraw from eculizumab, and we feel that the safety of this approach is a clinically more important endpoint.

5. What is tested in the biomarkers and complement activation sample? Does this include serum sC5b-9?

Yes, this will include soluble C5b-9, but will also include additional biomarkers of complement activation.

6. Have there been any other important modifications to the trial in light of the COVID-19 pandemic other than the study visits such as recruitment, trial pause, or other protocol modifications? If any, consider following this guideline:

<https://eur03.safelinks.protection.outlook.com/?url=https%3A%2F%2Fjamanetwork.com%2Fjournals%2Fjama%2Ffullarticle%2F2781397&data=04%7C01%7CSarah.Dunn2%40newcastle.ac.uk%7Cadb2ca033f4463b2dcd08da1c60529a%7C9c5012c9b61644c2a91766814f3e87%7C1%7C0%7C637853497278004866%7CUnknown%7CTWFPbGZsb3d8eyJWljoiMC4wLjAwMDAiLCJQIjoiV2luMzliLCJB TIl6lk1haWwiLCJXVCi6Mn0%3D%7C3000&sdata=Hj3SeGOXPJJ2x6c%2FurQEBaFnZvPd67bnZ4D PpOyQvkg%3D&reserved=0>

Thank you for providing this reference. Covid has had an effect on delivery of the study, and we refer to this impact in the manuscript (pages 3 and 7). More detail on the impact will be provided in the final report.

References

1. (Microsoft Word - NICE DSU TSD Survival analysis.updated March 2013 (nih.gov)
2. Introduction to health technology evaluation | NICE health technology evaluations: the manual | Guidance | NICE
3. Le Quintrec M, Zuber J, Moulin B, Kamar N, Jablonski M, Lionet A, et al. Complement genes strongly predict recurrence and graft outcome in adult renal transplant recipients with atypical hemolytic and uremic syndrome. *Am J Transplant.* 2013;13(3):663-75.
4. Ardissino G, Testa S, Possenti I, Tel F, Paglialonga F, Salardi S, et al. Discontinuation of eculizumab maintenance treatment for atypical hemolytic uremic syndrome: a report of 10 cases. *Am J Kidney Dis.* 2014;64(4):633-7.
5. <https://www.nice.org.uk/process/pmg36/chapter/introduction-to-health-technology-evaluation#medical-technologies-evaluation-programme>
6. Sheerin NS, Kavanagh D, Goodship TH, and Johnson S. A national specialized service in England for atypical haemolytic uraemic syndrome-the first year's experience. *QJM.* 2016;109(1):27-33.

VERSION 2 – REVIEW

REVIEWER	Alabdulqader, Muneera King Faisal University
REVIEW RETURNED	21-Jun-2022

GENERAL COMMENTS	Thank you for allowing me to review this protocol. It is comprehensive. and I have only minor suggestion 1- Table 1: Home Urinalysis result thresholds It should mention the Baseline for blood in urine: it is evident in the text, but it should also be mentioned in the table.
---

REVIEWER	Roberts, Matthew Monash University Eastern Health Clinical School
REVIEW RETURNED	10-Jun-2022

GENERAL COMMENTS	The authors have addressed all of my questions about the manuscript. This manuscript describes the study well.
--

REVIEWER	Yau , Kevin University of Toronto, Nephrology
REVIEW RETURNED	07-Jun-2022

GENERAL COMMENTS	The authors have satisfactorily answered all my queries and I have no further questions.
--

VERSION 2 – AUTHOR RESPONSE

Thank you for your comments and the opportunity to revise our protocol manuscript.
As requested by Reviewer 2, we have updated table 1. Table 1 now explicitly states that the urinalysis threshold relates to haematuria.